# Molecular motion of a nanoscopic moonlander via translations and rotations of triphenylphosphine on graphite
Anton Tamtögl [1] ✉, Marco Sacchi [2], Victoria Schwab [1], Michael M. Koza [3] & Peter Fouquet [3]

Mass transport at surfaces determines the kinetics of processes such as heterogeneous catalysis and thin-film growth, with the diffusivity being controlled by excitation across a translational barrier. Here, we use neutron spectroscopy to follow the nanoscopic motion of triphenylphosphine ($P(C_6H_5)_3$ or $PPh_3$) adsorbed on exfoliated graphite. Together with force-field molecular dynamics simulations, we show that the motion is similar to that of a molecular motor, i.e. $PPh_3$ rolls over the surface with an almost negligible activation energy for rotations and motion of the phenyl groups and a comparably small activation energy for translation. While rotations and intramolecular motion dominate up to about 300 K, the molecules follow an additional translational jump-motion across the surface from 350-500 K. The unique behaviour of $PPh_3$ is due to its three-point binding with the surface: Along with van der Waals corrected density functional theory calculations, we illustrate that the adsorption energy of $PPh_3$ increases considerably compared to molecules with flat adsorption geometry, yet the effective diffusion barrier for translational motion increases only slightly. We rationalise these results in terms of molecular symmetry, structure and contact angle, illustrating that the molecular degrees of freedom in larger molecules are intimately connected with the diffusivity.

Triphenylphosphine—$P(C_6H_5)_3$—is an important ligand for organic, organometallic, and nanoparticle synthesis[1,2] and shows a complex self-assembly behaviour on Au(111)[2]. However, in contrast to other organic compounds, the surface chemistry of $PPh_3$ has to date gone almost completely unexamined[3]. Here, we present an experimental and computational study of the diffusion of triphenylphosphine ($PPh_3$) on exfoliated graphite.

While the diffusion of polycyclic aromatic hydrocarbons (PAH) on graphite has been subject to several recent studies[4–7], the dynamics of other species on graphite is relatively unexplored. In general, surface diffusion in true thermal equilibrium has been mostly concerned with either very small molecules or flat adsorption geometries, as in the case of PAHs. Unlike previous studies of flat PAHs on graphite the $PPh_3$ molecule exhibits a completely different geometry: $PPh_3$ is pyramidal with a chiral propeller-like arrangement of the three phenyl rings (see Fig. 1(a))[8]. On hexagonal metal surfaces, $PPh_3$ molecules adsorb in an upright fashion in contrast to PAHs, which typically adsorb in a planar configuration. This adsorption geometry gives rise to a rather dense packing structure on the surface[2,9]. Hence in comparison to the dynamics of PAHs on graphite, the question arises, how the adsorption geometry influences the diffusive process.

Apart from the fundamental interest, the diffusion of $PPh_3$ on graphite is of great importance for applications. For example, phosphorus-doped graphene nanosheets can be prepared via annealing of graphene oxide in the presence of $PPh_3$. These nanosheets show excellent $NH_3$ sensing ability at room temperature[10,11]. Moreover, it was shown that the preparation of $PPh_3$-modified graphene quantum dots gives rise to a high quantum yield and excellent stability[12]. In a wider context, the diffusivity of triphenyl compounds is also of paramount importance for medical purposes: Triphenylbismuth and triphenylantimony are currently under investigation as contrast agents for magnetic resonance imaging[13,14] while triphenylphosphine has recently been incorporated in a macromolecule that can be used as a vehicle for mitochondrial drug delivery[15]. Relevant mobility data, in particular in the context of graphitic materials, is also of importance as similar phosphorous-containing molecules are frequently detected in natural environments as an emerging organic pollutant[16,17]. Triphenyl borate, on the other hand, has been introduced into lithium-ion batteries as a novel electrolyte additive and was found to improve the cycle and rate capability of the two anodes[18].

[1]Institute of Experimental Physics, Graz University of Technology, Graz, Austria. [2]Department of Chemistry, University of Surrey, GU2 7XH Guildford, UK. [3]Institut Laue-Langevin, 71 Avenue des Martyrs, 38000 Grenoble, France. ✉e-mail: tamtoegl@tugraz.at

**Fig. 1 | Adsorption geometry and energy landscape for PPh₃ on graphite. (a)** Molecular geometry of $P(C_6H_5)_3$ (PPh₃). **(b)** Different adsorption sites and rotational angles probed in the DFT calculations. **(c)** The energetically most favourable adsorption geometry of PPh₃ on graphite(0001) based on van der Waals corrected DFT calculations. PPh₃ adsorbs like a tripod, with the phenyl groups pointing towards the surface. The C-P angles of PPh₃ remain similar to the ones of PPh₃ in gas-phase but the phenyl groups tilt slightly towards a more planar configuration when the molecule is adsorbed. **(d)** Comparison of the potential energy surface as obtained by the vdW-corrected DFT for PPh₃ in the downward configuration for a rotation of 0°, 30° and 60°, respectively. The red and orange lines represent the first and second layers of the graphite substrate, respectively.

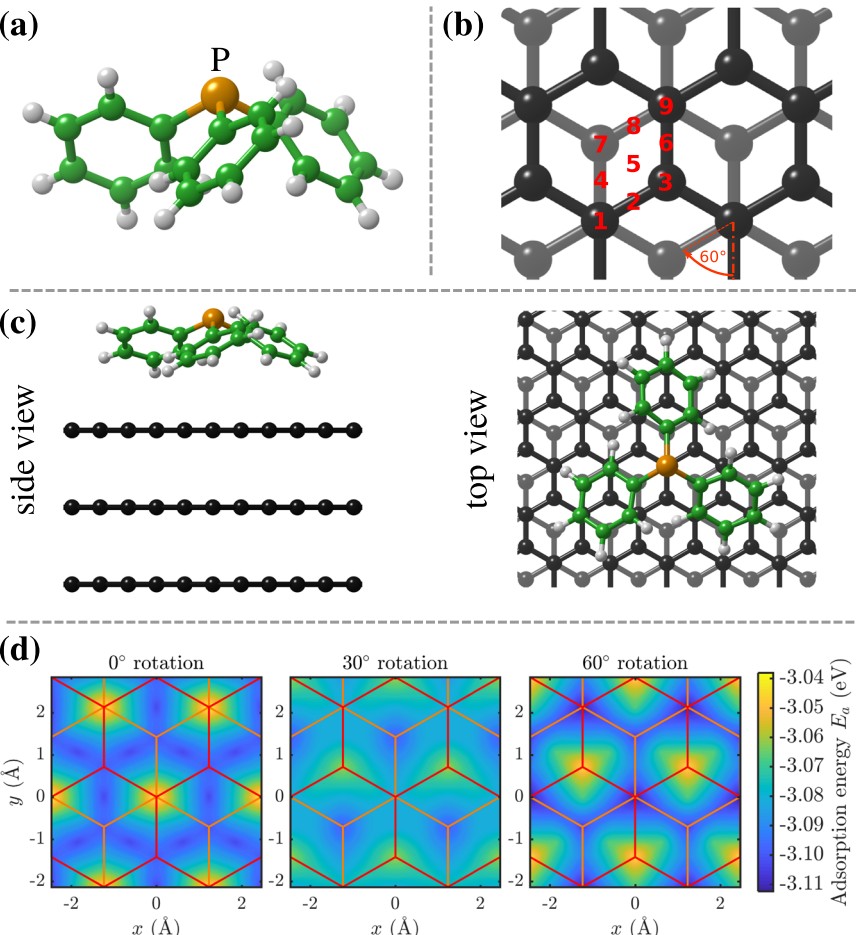

In this work we illustrate, based on neutron time-of-flight (TOF) and neutron spin-echo (NSE) measurements, that despite the large adsorption energy of PPh₃ the change of adsorption geometry from flat-lying molecules to a much smaller contact angle gives rise to a small diffusion barrier. The nanoscopic motion of PPh₃ adsorbed on exfoliated graphite at temperatures up to about 300 K is dominated by rotations of the molecule and flapping motion of the phenyl groups. With increasing temperature the dynamics is then dominated by translational motion with a relatively small activation energy from which we extract the diffusion coefficient for mass transport.

## Results and discussion
### DFT Results

Except for several studies on fcc metals, the adsorption and structure of PPh₃ on surfaces are relatively unexplored[3]. On fcc metal surfaces PPh₃ self-assembles to form highly ordered surface structures. Scanning tunnelling microscopy (STM) studies of PPh₃ on Au(111) show a well-ordered $(2\sqrt{3} \times 2\sqrt{3})R30°$ structure of PPh₃[2,19]. This particularly close packing arrangement for a rather large molecule is possible due to the upright adsorption of the PPh₃ molecules. Binding of PPh₃ to the surface occurs via the lone pair of the phosphorus atom with the phenyl groups oriented away from the surface, leading to a very symmetric adsorption geometry.

We have studied the adsorption of PPh₃ on graphite for a large number of different adsorption geometries. Those include 9 different adsorption sites within the graphite unit cell (labelled 1-9 in Fig. 1(b) and referred to the $C_3$ rotational axis through the P-atom), the orientation of the molecule with the phenyl groups pointing upwards (U) or downwards (D) as well as three different rotations around the axis perpendicular to the surface (with the P-C bond at an angle of 0°, 30° and 60° with respect to the y-axis).

The energetically most favourable adsorption geometry of (PPh₃) on graphite(0001) ($E_{ads}$ = 3.112 eV) based on vdW-corrected DFT calculations

is shown in Fig. 1(c). Note that the carbon atoms of PPh₃ are shown in green for illustrative purposes only. In general, PPh₃ adsorbs like a tripod, with the phenyl groups pointing towards the surface. The C-P angles of PPh₃ remain similar to the ones of PPh₃ in gas-phase but the phenyl groups tilt slightly towards a more planar configuration when the molecule is adsorbed (see Fig. 1(a,c)). The most favourable adsorption geometry is with the phosphorus atom directly on top of a carbon atom (site 9) and at a rotation of 60°.

Given the chemical similarity between PPh₃ and previously reported organic precursors, we expect that the accuracy of our calculations with PBE and TS corrections (see Computational details) will be comparable with our previous studies on graphitic and similar surfaces[4,5,20,21], on the order of 10 meV. Table 1 shows the adsorption energies $E_a$ for two different rotations in the downward configuration at the different adsorption sites (for the complete set of DFT calculations including all considered adsorption geometries please refer to the Supplementary DFT calculations and Supplementary Table 1). Compared with PAHs on graphite (benzene with 0.64 eV and pyrene with 1.56 eV[5]), the adsorption energy is quite large (around 3 eV), while the differences between the different positions and rotations are two orders of magnitude smaller. For PAHs on graphite, the binding energy per carbon atom has also been determined at ≈ 50 meV from thermal desorption spectroscopy[22] and thus, the adsorption energy of PAHs with similar size is clearly much smaller than for PPh₃. Compared to PPh₃ adsorption on Au, calculations predict adsorption energies varying between 3.0 and 4.7 eV depending on the truncation, while Jewell et al.[2] estimated an adsorption energy of about 1.3 eV on Au(111).

The most favourable adsorption sites are at a rotation of 60° with the P-atom located on top of a carbon atom that exhibits a second-layer carbon atom beneath (position 9 or 1). A translational motion between these two sites along a straight line would require crossing position 5 and correspond to a diffusion barrier of 46 meV based on the vdW DFT calculations. The

**Table 1 | The adsorption energy $E_a$ and the energy difference $\Delta E_a$ relative to the most favourable adsorption site for PPh$_3$ adsorption on graphite with the phenyl groups pointing towards the surface**

| Pos. | 60° Rotation | | 0° Rotation | |
|------|--------------|--------------|--------------|--------------|
| | $E_a$ (eV) | $\Delta E_a$ (meV) | $E_a$ (eV) | $\Delta E_a$ (meV) |
| 1 | -3.111 | 1 | -3.042 | 70 |
| 2 | -3.078 | 34 | -3.075 | 37 |
| 3 | -3.043 | 69 | -3.099 | 13 |
| 4 | -3.092 | 20 | -3.081 | 31 |
| 5 | -3.066 | 46 | -3.102 | 10 |
| 6 | -3.084 | 28 | -3.081 | 32 |
| 7 | -3.104 | 9 | -3.100 | 12 |
| 8 | -3.100 | 13 | -3.073 | 39 |
| 9 | -3.112 | 0 | -3.038 | 74 |

The 9 different adsorption sites within the graphite unit cell (positions 1–9 in Fig. 1(b)) are referred to the C$_3$ rotational axis through the P-atom.

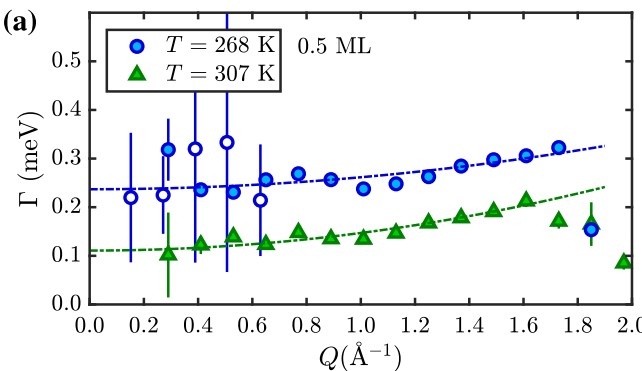

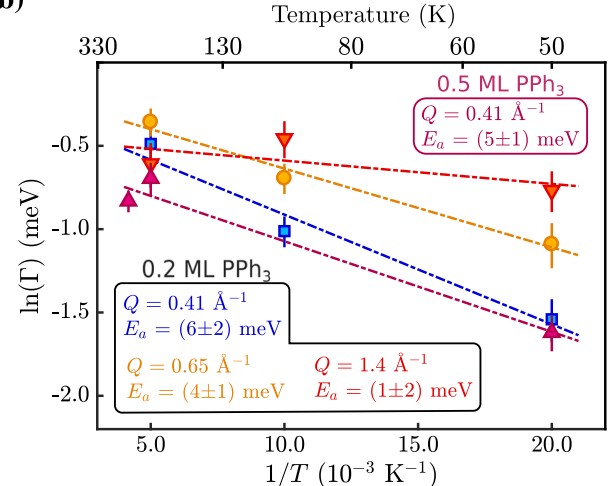

**Fig. 2 | Low-temperature dynamics of PPh$_3$ on graphite. (a)** Already at temperatures below room temperature, a quasi-elastic broadening $\Gamma$ is present in both NSE (open symbols) and neutron TOF data (closed symbols) for PPh$_3$/graphite (0.5 ML). The functional dependence $\Gamma(Q)$ with the dotted lines is for uniaxial rotations and suggests that dynamics is dominated by molecular rotation and motion of the phenyl groups (see text). **(b)** An Arrhenius plot of $\Gamma$ in the 50–300 K temperature regime illustrates that for both the 0.2 and 0.5 ML coverage TOF data with $E_a \approx 5$ meV no significant barrier for this type of dynamic motion exists. The error bars correspond to the confidence bounds ($1\sigma$) upon determination of $\Gamma$ from the measurements (see text).

barrier for a rotation of the molecule around the C$_3$ axis depends strongly on the adsorption position and is as small as 3−4 meV for adsorption on position 2, 6 or 7.

On the other hand, an inversion of the whole molecule from the downward to the upward configuration, assuming that the molecule stays in the same rotation and position, requires much more energy, between about 100 meV and 200 meV depending on the actual position. Hence, this motion is rather unlikely to be observable at the experimental temperatures (≤500 K) of this study.

Moreover, while "flat" PAHs do not exhibit a dipole moment, PPh$_3$ has a dipole moment of about 1.4 Debye[1], similar to water. Due to this dipole moment, possible lateral interactions between the adsorbates may play an important role during diffusion and self-assembly of PPh$_3$. Interactions among the adsorbates, such as dipole forces, give rise to a deviation in the QENS broadening as a function of momentum transfer **Q**. Such a behaviour has been predicted and described using analytical models already in 1959 by de Gennes[23]. However, while it has been studied and observed in liquids, the role of adsorbate interactions for the diffusion on surfaces has been hardly covered by experiments[24–27].

## Experimental results

The experimentally measured scattering function $S(Q, \Delta E)$ (normalised by vanadium) was fitted using a convolution of the resolution function of the neutron TOF spectrometer $S_{res}(Q, \Delta E)$ (scattering function of PPh$_3$/graphite measured at ≈ 2 K) with an elastic term $I_{el}(Q)\delta(\Delta E)$, the quasi-elastic contribution $S_{inc}(Q, \Delta E)$ and a constant background:

$$S(Q, \Delta E) = y_0 + S_{res}(Q, \Delta E) \otimes \left[ I_{el}(Q)\delta(\Delta E) + A(Q)\frac{1}{2\pi}\frac{\Gamma(Q)}{[\Gamma(Q)]^2 + \Delta E^2} \right]. \quad (1)$$

Here, $\delta$ represents the Dirac delta, and the quasi-elastic broadening is modelled by a Lorentzian function, where $I_{el}(Q)$ is the intensity of the elastic scattering, and $A(Q)$ is the intensity of the quasi-elastic scattering. $\Gamma(Q)$ is the half-width at half maximum (HWHM) of the Lorentzian.

The intermediate-scattering function (ISF) obtained from the spin-echo measurements was fitted using a stretched exponential decay function (the so-called Kohlrausch Williams Watts function, KWW)

$$I(Q, t) = (1 - y_0)\exp\left[-\left(\frac{t}{\tau_{WW}}\right)^{\beta}\right] + y_0 \quad (2)$$

where $y_0$ is the fraction of static signal and $\beta$ is the stretching exponent ($\beta \le 1$, with $\beta = 1$ for a single exponential decay). For comparison with the neutron TOF data, the obtained decay $\tau_{WW}$ can be related to the broadening in the energy domain via $\Gamma = \hbar/\tau_{WW}$.

Exemplary TOF and NSE fits are shown in Supplementary Figs. 1, 2, and all plotted error bars correspond to $1\sigma$ confidence bounds. Hydrogenated PPh$_3$ is a strongly incoherent scatterer, and thus, the TOF measurements are strongly dominated by incoherent scattering and probe the diffusion of single protons in the molecules. By using deuterated triphenylphosphine (P(C$_6$D$_5$)$_3$) in the NSE measurements, we can switch to a mostly coherent scatterer in the low $Q$ range. Hence, TOF measurements are sensitive to the single molecule averaged trajectories, while the coherent scattering function obtained in NSE measurements corresponds to the averaged relative molecular trajectories in the collective system.

## Low-temperature dynamics

As mentioned above, according to vdW-corrected DFT calculations, the adsorption energies of PPh$_3$ (3.11 eV) compared to other molecules with flat adsorption geometry and similar mass such as pyrene (C$_{16}$H$_{10}$, $E_a = 1.56$ eV according to Ref [5]) are much larger, while the energy differences between several adsorption sites (Table 1) remain small, i.e., in the region of 10s of meV. In fact, both NSE and neutron TOF data show a broadening $\Gamma$ (Fig. 2)

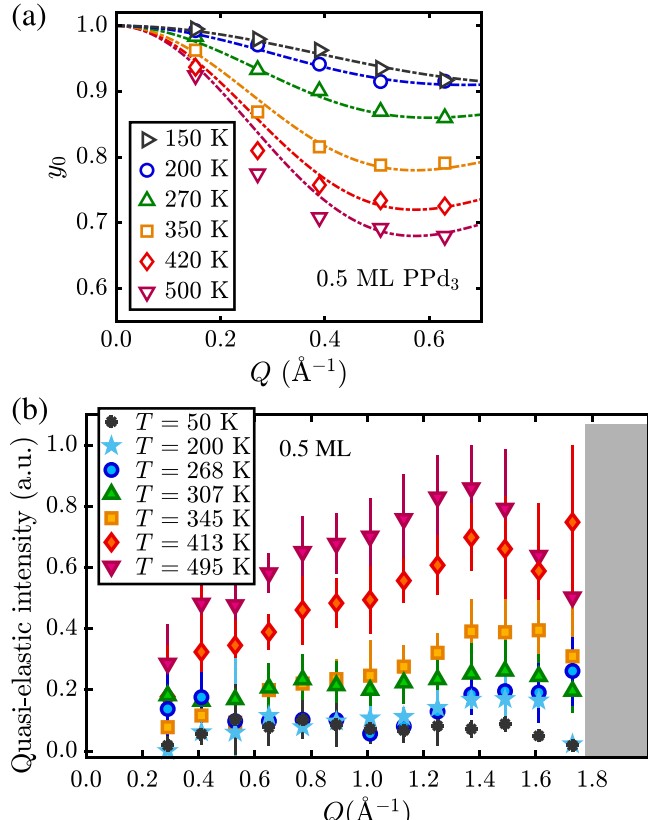

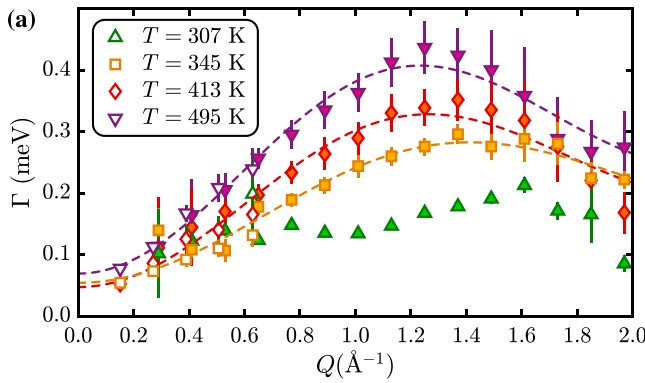

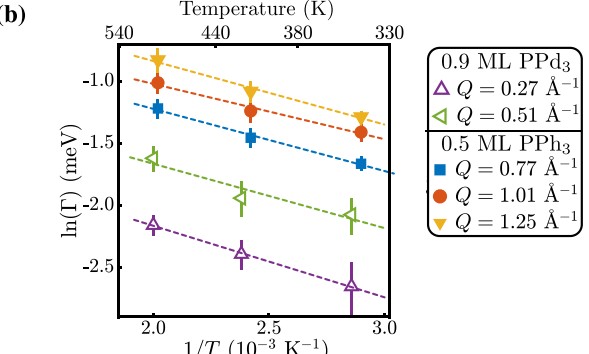

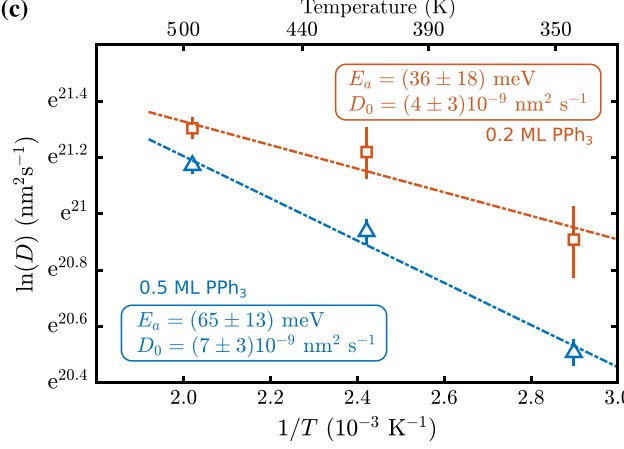

**Fig. 3 | Elastic and quasi-elastic neutron scattering intensities over the entire temperature range.** As can be seen in (**a**), $y_0$ from the NSE data illustrates that the low-temperature dynamics is dominated by rotation and motion of the phenyl groups. The decrease of $y_0$ with $Q$ becomes more pronounced with increasing temperature and is well described by Eq. (4) (dashed–dotted lines) up to about 350 K, thus corresponding to an increasing contribution from rotational dynamics with a radius that equals roughly the size of a phenyl group. The quality of the fit decreases above 350 K because here, translational motion starts to dominate. The quasi-elastic intensities from the TOF data in (**b**) show that the quasi-elastic component increases continuously with temperature (error bars correspond to $1\sigma$ confidence bounds). The grey-shaded region corresponds to areas around the Bragg peak of graphite, which is dominated by coherent scattering of the carbon atoms of the substrate and thus makes extraction of the quasi-elastic amplitude difficult.

**Fig. 4 | High-temperature dynamics and diffusion parameters for PPh₃ on graphite.** (**a**) The extracted quasi-elastic broadening $\Gamma(Q)$ for 0.5 ML PPh₃ in the high-temperature regime versus momentum transfer $Q$, illustrates that the motion turns into a translational jump-diffusion. From about 350 K onward, the data points (open symbols are NSE and filled symbols TOF data) clearly follow the dashed lines according to Eq. (5), i.e. the functional dependence for jump diffusion. (**b**) From an Arrhenius plot at several fixed momentum transfers $Q$, a diffusion barrier of $E_a = (44 \pm 4)$ meV is extracted for the activated hopping motion in the $350-500$ K temperature range (0.5 ML TOF and 0.9 ML NSE data). (**c**) Illustrates the parameters for mass transport, i.e. the diffusion coefficients $D$ as extracted from the fits of Eq. (5). The 0.5 ML data follows directly from panel (a), while the fits for the 0.2 ML data are shown in Supplementary Fig. 4. All error bars correspond to $1\sigma$ confidence bounds.

and quasielastic intensity (Fig. 3) occurring already at low temperatures and at all three coverages (0.2, 0.5 and 0.9 monolayers (ML)) that were studied. As can be seen in Figs. 2(a), 4(a) the broadening $\Gamma(Q)$ does not approach zero with $Q \rightarrow 0$. Moreover, $\Gamma(Q)$ in Fig. 2(a) does not show a clear functional dependence with respect to the substrate lattice, except for a few points close to the graphite diffraction peak where the quasi-elastic signal becomes "shadowed" by the elastic graphite peak.

The dynamics of small molecules such as methane and ethane on graphite have been studied previously with neutron scattering[28–31] showing similarities to the here observed low-temperature motion. As described by Thorel et al.[31], the broadening for isotropic rotational diffusion of methane on graphite is almost $Q$-independent. In general, for PPh₃ the overall trend for $\Gamma(Q)$ upon approaching room temperature follows uniaxial rotations, where in the current setup, a constant offset and a slow increase of $\Gamma$ with $Q^2$ is expected[32]. We note that this is, of course, a very simplified model, but without better data and other analytic solutions for a complex molecule such as PPh₃, the broadening can be described as $\Gamma = D_\perp \cdot q_\perp^2 + D_\parallel \cdot Q_\parallel^2$ with the parallel (∥) and perpendicular (⊥) component with respect to the basal plane of graphite. We are only sensitive to $Q_\parallel$ in the current scattering geometry and obtain $D_\parallel = 1.9 \times 10^{-10}$ m² s⁻¹ at 270 K and $D_\parallel = 2.8 \times 10^{-10}$ m² s⁻¹ at around 300 K, compared to isotropic rotational diffusion of

methane on graphite with $D_r = 6 \times 10^{-11}$ m² s⁻¹, with the latter being however measured at 55 K[31]. Hence, the low-temperature phase is likely to be dominated by rotational motion[20] as well as perpendicular motion such as "flapping" of the phenyl groups[20,24] (see Supplementary video 1 and 2 from the MD simulations). Only at higher temperatures, starting from about 350 K, diffusion is dominated by translational motion, as further described below.

The dynamics at low temperature is further characterised by an almost negligible activation energy $E_a$ as illustrated in the Arrhenius plot in

Fig. 2(b). According to

$$\Gamma = \Gamma_0 \, \exp\left(-\frac{E_a}{k_B T}\right), \qquad (3)$$

with the activation energy $E_a$, we obtain $E_a \approx (5 \pm 2)$ meV from Fig. 2(b) in the 50−300 K temperature regime and thus no significant barrier for this type of dynamic motion is present. We note that the uncertainty of the broadening extracted from the neutron TOF data in the low-temperature regime below 270 K is relatively large, as shown in Supplementary Fig. 3(a). Consequently, the data is quite scattered over the $Q$-range, and there is also some variation in the $E_a$ values calculated at different $Q$ in Fig. 2(b), meaning that $E_a \approx 5$ meV can be seen as an upper limit to the low-temperature motion.

Moreover, further evidence for the described type of motion comes from the NSE data. As it is difficult to extract the actual elastic incoherent structure factor (EISF) due to the uncertainties of the fits in the low-temperature regime, we consider the elastic amplitude $y_0$ of the NSE data according to Eq. (2). As shown in Fig. 3(a) and Supplementary Fig. 3(b), the elastic amplitude $y_0$ illustrates that an inelastic contribution is already present at 150 K which further increases with increasing temperature. In analogy to the $Q$ dependence of an EISF, the functional dependence of $y_0$ versus momentum transfer $Q$ for simple uniaxial rotations follows from:

$$y_0(Q) = (1 - A) \cdot j_0^2(Q \cdot \rho) + A \qquad (4)$$

where $j_0$ is the zeroth-order Bessel function, $\rho$ is the radius for the rotations[33] and $A$ corresponds to the remaining purely elastic signal. Indeed, the low-temperature data in Fig. 3(a), fitted with Eq. (4) corresponds to $\rho \approx 3-4$ Å, while the radius from one of the outer H-atoms to the central P-atom is roughly 4.2 Å.

## High-temperature motion

A striking change in motion and activation energy of the dynamical process is observed at approximately 300 K. The motion at 350−500 K can be attributed to translational jumps as described below, following a "rolling" motion, whereas very weakly activated rotation and motion of the phenyl groups dominate up to about 350 K. Both the low-temperature broadening from rotations and the high-temperature translational motion appear on similar timescales, but once the barrier for the translational motion is overcome, the latter will dominate the broadening[20,34].

While the intensity of the quasielastic broadening increases continuously with temperature, as shown in Fig. 3(b), it gives rise to the rather unusual fact that the broadening in the transition region first seems to decrease from 270 K to $\approx 300$ K before eventually increasing again continuously in the high-temperature window where the translational motion dominates. Evidence for this change of motion also comes from the NSE fits in Supplementary Fig. 2, where the ISF is better fitted with a stretched exponential function, in particular in the temperature region where the transition to the translational motion occurs[35].

Rotational and translational processes occur concurrently with different levels of thermal activation, as shown in previous spin-echo studies of molecular surface motion[20,34]. This has also been shown for rotational motion in the sense of internal motion in more complex systems[36,37]. Consequently, the weak momentum transfer dependence of the broadening $\Gamma(Q)$ in Fig. 4(a) around room temperature turns into a clear functional dependence versus $Q$ with increasing temperature up to about 500 K. The diffusion of the adsorbate becomes dominated by the interaction of the molecule with a corrugated surface, and its motion can be well described by an analytical model of jump diffusion[38,39]. In the case of scattering from a polycrystalline sample, isotropic angular averaging has to be performed since the scattered neutron signal "sees" the jumping adsorbate from all possible directions. In the case of 2D isotropy, integration in the scattering plane (over the azimuth $\varphi$) yields:

$$\Gamma(Q) = \frac{\hbar}{\tau}\left[1 - J_0(Q \cdot l \cdot \sin\vartheta)\right], \qquad (5)$$

where $J_0(Q \cdot l \cdot \sin\vartheta)$ is the zeroth order cylindrical Bessel function, $l$ is the average jump length and $\tau$ is the residence time of the molecule between two consecutive jumps. $Q \cdot \sin\vartheta$ is the component of the scattering vector in the plane of diffusion, and $\vartheta$ the angle between $\mathbf{Q}$ and the normal to the plane[40].

Papyex consists of planes with an inclination that is normally distributed around $\vartheta = 90°$ with a HWHM of about 15°[41], which has been taken into account by numerical integration of Eq. (5)[42]. Equation (5) is then fitted to the experimentally determined broadening $\Gamma(Q)$ using an iterative generalised least squares algorithm with weights. Moreover, a constant offset is added to Eq. (5) as an additional fit parameter to account for the perpendicular motion (rotation and motion of the phenyl groups), which also contributes to the broadening. The dashed lines in Fig. 4(b) show that Eq. (5) fits the data very well for an average jump length of $(1.1-1.3)a_{gr}$ with increasing temperature (see Table 2, with $a_{gr}$ being the in-plane graphite lattice constant). From the momentum transfer dependence, we can thus clearly attribute the high-temperature motion to hopping or jump diffusion.

Other types of motion, such as ballistic diffusion, would be characterised by a linear dependence of $\Gamma(Q)$, and Brownian or continuous motion is characterised by a square law dependence. As described in Low-temperature dynamics, intramolecular motion, i.e. rotations of the molecule and the motion of the phenyl groups at low temperature, gradually turns into diffusive, i.e. translational motion across the surface with increasing temperature.

The Arrhenius plot in Fig. 4(b) further illustrates that for the 350−500 K data a clear slope and hence an activated process is present, in contrast to the low-temperature regime displayed in Fig. 2(b). The slopes are consistent for both the 0.5 ML neutron TOF and the 0.9 ML NSE data. Based on the average over various $Q$ values, we obtain an activation energy of $E_a = (44 \pm 4)$ meV for the translational jump-diffusion. The experimental value is in very good agreement with vdW-corrected DFT calculations in DFT Results, which predict a translational barrier of 46 meV. The latter is of course obtained from a 0 K calculation, assuming straight motion between the sites and can, thus, only serve as an estimate.

Our interpretation of the molecular motion of $PPh_3$ based on the neutron scattering measurements is further confirmed by force field molecular dynamics (MD) simulations. MD simulations provide additional insight and interpretation into the real-space motion of the molecules beyond analytical models and have been shown to reproduce similar systems[43] accurately. Supplementary video 2 shown as extracted from the MD simulations for a single $PPh_3$ molecule on graphite at 300 K and Supplementary video 1 and 3 for 0.5 ML of $PPh_3$ at 50 K and 500 K, respectively, illustrate that the slow translational motion is accompanied by molecular rotations and motion of the phenyl groups resulting in a "rolling" motion similar to a nanoscopic moonlander. An analysis of the MD simulations for 0.5 ML of $PPh_3$ on graphite is shown in Fig. 5 (for details, please refer to the Supplementary details of the force field MD simulations).

The broadening $\Gamma(Q)$ extracted from fits to the ISFs shows good agreement with the experimental data (triangles) in the small to medium $Q$-range, i.e. at medium to long real-space distances. It indicates that the long-range motion of the molecules is well represented in the simulation. However, at higher $Q$-values (from $\approx 1.4$ Å$^{-1}$), the broadenings extracted

**Table 2 | Diffusion parameters according to fitting Eq. (5) to the 0.5 ML neutron TOF and NSE data (Fig. 4(a)) in the 350–500 K temperature range**

| $T$ (K) | $l$ (Å) | $\tau$ (ps) | $D$ (m² s⁻¹) |
|---|---|---|---|
| 345 | 2.8 ± 0.04 | 4.0 ± 0.1 | $(8.1 \pm 0.4) \times 10^{-10}$ |
| 413 | 3.13 ± 0.04 | 3.3 ± 0.1 | $(12.4 \pm 0.6) \times 10^{-10}$ |
| 495 | 3.21 ± 0.02 | 2.72 ± 0.04 | $(15.7 \pm 0.5) \times 10^{-10}$ |

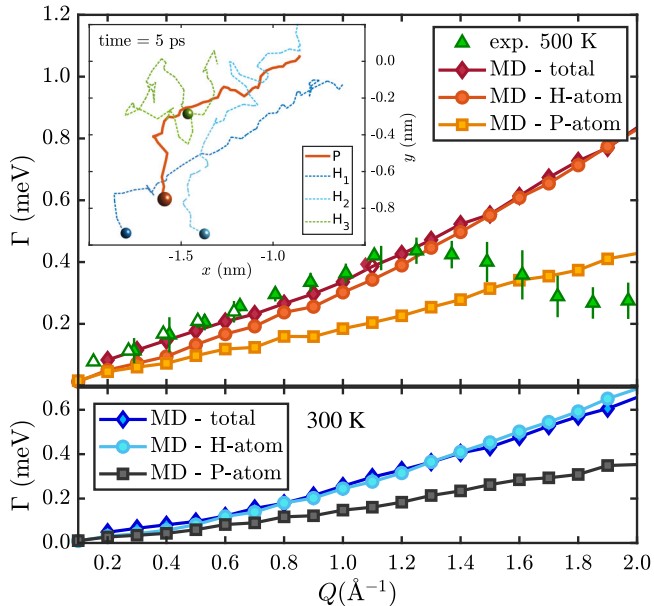

**Fig. 5 | Broadening Γ extracted from molecular dynamics simulations, together with an exemplary trajectory.** Broadening Γ extracted from calculated ISFs according to force field MD trajectories of 0.5 ML PPh$_3$ at 300 K (bottom panel) and 500 K (top panel) together with the experimental data (triangles). Γ(Q) obtained from the ISFs of the total system as well as the H-atoms shows good agreement with the experimental data at medium to long-range real-space distances but deviates significantly for the short-range motion (large Q, see text). Moreover, both plots clearly show that Γ(Q) obtained from the P-atom trajectories gives rise to a much smaller broadening at large Q. The inset in the top-left corner shows a 5 ps snapshot of one MD trajectory for the central P-atom and one H-atom of each phenyl group.

from the MD simulation deviate significantly from the experimental values, following a linear to slightly quadratic Q-dependence. Since the functional behaviour of the experimental values can be assigned to analytical jump diffusion according to Eq. (5), this type of motion is clearly not correctly described by the MD simulation. The experimental data shows that between two consecutive jumps, i.e. at real-space distances where the molecule approaches the next adsorption well, the rate is slowed down resulting in a dip at $\approx 2$ Å$^{-1}$. However, as can be seen from Supplementary video 3, in the high-temperature regime, the PPh$_3$ molecules desorb rather quickly from the surface, giving rise to a short interaction in terms of the underlying corrugated potential energy surface between the molecule and the graphite substrate. Consequently, the hopping motion between lattice sites that is characteristic of jump-diffusion is not accurately represented. It may indicate that the adsorption energy for tripod-like molecules such as PPh$_3$ with graphite is underestimated by the force field.

In addition to the ISFs calculated from all-atom trajectories of the MD simulation, we also calculated the ISFs solely for the P-atoms and the H-atoms in the system in order to get insight into the intra- and intermolecular types of motion (see Supplementary Fig. 6). Γ(Q) obtained from the ISFs for the P-atom trajectories can be associated with the centre-of-mass motion in the surface plane. The motion of H-atoms, on the other hand, can be linked to the additional molecular degrees of freedom. As can be seen from Fig. 5, with increasing Q, the values of Γ become much larger for the total system and the H-atoms compared to the P-atoms, illustrating that the H-atoms follow a faster dynamical behaviour. The latter is also evident from the 5 ps snapshot of a single PPh$_3$ MD trajectory shown in Fig. 5: While the trajectory of the central P-atom is more "straight-foward", the trajectories of the phenyl groups, represented by one hydrogen-atom per group, show a much more intricate motion.

As can be seen from Supplementary Fig. 7, the activation energy $E_a$ can also be determined from the MD simulations yielding for the low-

temperature motion from 50 to 300 K, $E_a \approx (3.8 \pm 0.5)$ meV, in good agreement with the experimentally determined value. It confirms the small energy barrier for uniaxial and intramolecular rotation found in the experiment, despite the limitations of the MD simulation in the high-temperature regime and the discrepancy at larger Q as further discussed in Supplementary details of the force field MD simulations.

### Diffusion parameters for PPh$_3$ on graphite

Moreover, based on the fits of Eq. (5) to the experimental neutron TOF and NSE data in the $350 - 500$ K temperature range at 0.5 ML coverage in Fig. 4(a), we can further extract the diffusion parameters for mass transport of PPh$_3$ on graphite, as summarised in Table 2 and plotted in the Arrhenius plot of Fig. 4(c). The diffusion coefficient, $D$, for two-dimensional mass transport is therefore calculated using $D = \frac{1}{4\tau}\langle l\rangle^2$, with the residence time, $\tau$, and the average jump length, $\langle l\rangle$, being derived from the fits of Eq. (5) to the experimental broadening. We note that the barrier extracted from Fig. 4(c) is slightly larger compared to the Arrhenius' at a fixed Q-value but also exhibits a larger uncertainty. From the extracted diffusion coefficients $D$ we obtain $D_0 = (7 \pm 3) \times 10^{-9}$ m$^2$ s$^{-1}$. Previous experimental reports about dynamics include NMR relaxation times of triphenylphosphine oxide adsorbed on aluminium[44] and the formation of clusters in real space[45]. However, molecular diffusivities and diffusion coefficients have not been available up until now. Compared to the diffusion of small molecules on graphite[27,42], the translational component of the overall motion that contributes to the surface diffusion of PPh$_3$ is smaller. For larger molecules, e.g., the prototypical benzene/graphite system, a diffusion coefficient of $5.4 \cdot 10^{-9}$ m$^2$/s was reported at 140 K by Hedgeland et al.[7]. Similarly, for orientation-dependent diffusion of pentacene molecules on a pentacene layer at 300 K a much higher diffusivity was reported based on simulation results[46]. On the one hand such large diffusivities may be the consequence of an unusual low friction of PAHs on graphitic substrates[47,48]. On the other hand, for cobalt phthalocyanine, another planar molecule with about twice the mass compared to PPh$_3$, a larger activation energy but a similar diffusivity with $9.6 \times 10^{-10}$ m$^2$ s$^{-1}$ at 350 K was found on metal surfaces[49]. In the latter case it was noted that the additional molecular degrees such as rotations may in turn increase the diffusivity of large organic molecules. Hence for PPh$_3$ the different symmetry and increased complexity of the molecule may reduce the frictional coupling to the surface during translational motion while at the same time the additional degrees of freedom such as rotations and motion of the phenyl groups provide further energy dissipation channels.

Finally, we shortly discuss the possible influences of the PPh$_3$ coverage on the dynamics and mass transport. The same analysis as for the 0.5 ML sample can be performed for the 0.2 ML TOF data, also plotted in Fig. 4(c), where the detailed values can be found in Supplementary Table 2. While the uncertainties are larger than for the higher coverage, as seen from the figure, $D$ is slightly larger for 0.2 ML compared to 0.5 ML. Together with a more detailed discussion in Supplementary coverage dependence analysis, we conclude that with increasing coverage, the translational motion is slowed down to some extent, possibly via repulsive interactions or simply via site blocking. At the same time, the rotational motion becomes more important with increasing coverage, likely due to increased confinement upon more molecules being present at the surface (see Supplementary coverage dependence analysis). However, the activation energy for $E_a$ appears to be unaffected by the coverage, at least within the uncertainties, as can be seen both in Fig. 4(c, b).

### Conclusions

In summary, we have studied the dynamics of triphenylphosphine (P(C$_6$H$_5$)$_3$ or PPh$_3$) adsorbed on exfoliated graphite. In contrast to flat polycyclic hydrocarbons, which have been studied widely in recent years, PPh$_3$ shows a three-dimensional adsorption geometry similar to a nanoscopic moonlander. Quasi-elastic neutron scattering measurements of PPh$_3$ illustrate that the surface dynamics of large non-planar adsorbates give rise to a more complex diffusion mechanism. We have observed that by

deviating from simple 2D adsorption geometries, we find an almost negligible barrier for molecular rotations and flapping motion of the phenyl groups, while the activation energy for translations at 44 meV, albeit not negligible, remains relatively small despite a considerable increase of the adsorption energy. Hence, as a consequence of the change of adsorption geometry compared to flat (face-face) configurations, a much larger adsorption energy of $PPh_3$ is obtained from vdW-corrected DFT, while the relatively small activation energy for translations is likely related to a much smaller contact angle of $PPh_3$.

The change of molecular and adsorption geometry leads to a two-fold difference in the dynamics of $PPh_3$: The molecular structure allows for molecular rotation and flapping of the phenyl groups, which are already present at low temperatures, as also seen in molecular dynamics simulations. At the same time, an increase of adsorption energy compared to other molecules with flat adsorption geometry and similar mass means that the translational motion starts to set in at higher temperatures, while once the translational dynamics sets in, the diffusion barrier remains small.

In particular, for more complex molecules, we are only now starting to gain a better fundamental understanding of how the molecular degrees of freedom contribute to mass transport and the factors controlling thin film growth and heterogeneous catalysis[49]. While traditionally, the motion of simple molecules is often treated as point-like particles, molecular degrees of freedom can influence diffusion dynamics significantly, and rotations already play a role for simple molecules such as benzene on metal substrates[50]. In larger molecules, rotations and conformational changes are intimately connected to translational motion guiding the molecule towards the lowest barrier of the energy landscape and thus, internal coordinates can accelerate the crossing of energy barriers[49]. Our study thus highlights the importance of exploring the full potential energy landscape and provides benchmark data as we seek to develop models that predict the rates of chemical reactions, covering dynamics at high temperatures and allowing a precise determination of diffusion coefficients and the pre-exponential factor.

## Methods
### Experimental details
**Sample preparation.** As a substrate, we used exfoliated compressed graphite, *Papyex*, which exhibits an effective surface area of about 25 m² g⁻¹ and retains a sufficiently low defect density[41,51]. Due to its high specific adsorption surface area, it is widely used for adsorption measurements[52]. We further exploit the fact that exfoliated graphite samples exhibit a preferential orientation of the basal plane surfaces, and we oriented those parallel to the scattering plane of the neutrons. Each sample was prepared with $12-13$ g of Papyex exfoliated graphite of grade N998 ( > 99.8% C, Carbone Lorraine, Gennevilliers, France). The prepared exfoliated graphite disks were heated to 973 K under vacuum before being transferred into a cylindrical aluminium sample cartridge. The amount of powder $PPh_3$ ($C_{18}H_{15}P$ and $C_{18}D_{15}P$, respectively), required to reach the corresponding ML coverage, was weighed using a fine balance and then added to the graphite disks. At monolayer coverage, the area occupied by one $PPh_3$ molecule corresponds to $\Sigma = 63$ Å² (based on Ref. [2]). Finally, the aluminium sample holders were hermetically sealed using a lid with a steel knife edge. The samples were then heated in a furnace to 400 °C to sublimate the $PPh_3$ and promote its adsorption in the whole volume of the sample, in analogy to Refs. 4,5.

**Instrumental details.** The measurements were carried out at the IN6 time-of-flight (TOF) neutron spectrometer and the IN11 neutron spin-echo (NSE) spectrometer of the ILL[53]. The incoming neutron wavelengths were set to 5.12 Å and 5.5 Å, respectively, with energy resolutions at full width at half maximum of 70 μeV (IN6) and 1 μeV (IN11). Neutron scattering TOF spectra of $P(C_6H_5)_3$/graphite were obtained over a large temperature region, ranging from 2 K to 500 K (at 0.2 ML and 0.5 ML $PPh_3$ coverages, respectively). Neutron spin-echo measurements of $P(C_6D_5)_3$/graphite were performed over the same thermal range for 0.5

ML and 0.9 ML coverages, respectively. For molecules such as $PPh_3$, there exists a pronounced contrast between the scattering cross-sections of hydrogen and carbon[39], which allows us to distinguish between the signal arising from the adlayer and from the substrate.

As further detailed in Supplementary methods, the TOF spectra were converted to scattering functions, $S(Q, \Delta E)$, where $Q = |\mathbf{Q}| = |\mathbf{k_f} - \mathbf{k_i}|$ is the momentum transfer and $\Delta E = E_f - E_i$ is the energy transfer. As can be seen in Supplementary Fig. 1(a), for the neutron TOF data, areas of low momentum transfer, i.e. $Q \le 0.2$ Å⁻¹ and around the Bragg peak of graphite at $Q \approx 1.8$ Å⁻¹ are dominated by coherent scattering of the carbon atoms of the substrate.

NSE measurements, on the other hand, deliver the development of the space correlation function with time $t$, i.e., the normalised intermediate scattering function $I(Q, t)/I(Q, 0)$[54,55]. The intermediate scattering function is related to the scattering function $S(Q, \Delta E)$ via a Fourier transform in time.

### Computational details
DFT calculations were performed using the plane wave periodic boundary condition code CASTEP[56]. We have employed the Perdew-Burke-Ernzerhof (PBE)[57] exchange-correlation functional, with the dispersion force corrections developed by Tkatchenko and Scheffler (TS method)[58] for the calculations presented in this work. The plane wave basis set was truncated to a kinetic energy cutoff of 360 eV. The adsorbate system was modelled using a $(9 \times 9)$ graphene unit cell composed of a three-layer graphene sheet and a vacuum spacing of 20 Å above the graphite surface in order to avoid interactions with the periodically repeated supercells.

### Force field molecular dynamics simulations
Force field molecular dynamics simulations were performed using the force field engine Forcite within the program package Materials Studio[59] (see Supplementary Fig. 5 for the setup of the system). We used the thermodynamic ensemble NVT (constant particle number, volume and temperature) employing a Nosé-Hover thermostat. The dynamics were calculated for 1 fs time steps to include the fast hydrogen dynamics and the atomic positions were written into the trajectory files every 20 fs. The simulated trajectory files were further converted to the intermediate scattering function (ISF) $I(Q, t)$ and analysed using the Python package MDANSE[60]. Further details can be found in the supplementary details of the force field MD simulations.

## Data availability
The raw neutron scattering data is available from ref. [53]. The vdW-corrected DFT data supporting the conclusions of this study are available via the TU Graz repository.

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

## Acknowledgements

This research was funded in whole, or in part, by the Austrian Science Fund (FWF) [P29641-N36 & P34704]. For the purpose of open access, the author has applied a CC BY public copyright licence to any Author Accepted Manuscript version arising from this submission. M.S. is grateful for the support from the Royal Society (URF\R\191029). This work used the ARCHER UK National Supercomputing Service via the membership of the UK's HEC Materials Chemistry Consortium, funded by the EPSRC (EP/X035859). The authors acknowledge the generous provision of neutron beam time at the ILL.

## Author contributions

A.T. prepared, carried out the neutron scattering measurements and wrote the first version of the manuscript. M.S. performed the vdW-corrected DFT calculations. V.S. carried the force field MD simulations and analysis out. P.F. and M.M.K. evaluated the data together with A.T. All authors contributed to the discussion and interpretation of the results and approved the final version of the manuscript.

## Competing interests

The authors declare no competing interests.
