## [Peer Review File · Communications Chemistry]

Reviewers' comments:

Reviewer #1 (Remarks to the Author):

In their manuscript "Mobility of a Nanoscopic Moonlander: Translations and Rotations of Triphenylphosphine on Graphite" the authors present combined neutron spin echo, time of flight and DFT study of the dynamics of PPh₃ molecules on a graphite surface. The theoretical and experimental results agree well and the conclusions drawn are supported by the data presented. Overall, this presents a solid study of 2D diffusion on surfaces. This results is very interesting and shows how molecular rotations present at the lower temperatures investigated turn into lateral diffusion at higher temperatures. As stated by the authors the work extend existing literature on planar adsorbents to more complex systems. In the introduction the authors underline the importance of their findings with respect to the surface diffusion but they do not say how their results measuring the diffusion on graphite surfaces are related to those cases. Overall, I judge the manuscript very positive and support publication in Communications Chemistry.

Below some minor points, which the authors should clarify prior to publication:

Low temperature dynamics: The authors discuss in the text different surface coverages but data is only shown for 0.5 ML in the manuscript. Some of the other data is shown in the supporting material but not all coverages mentioned. The authors should show the data to support their statements.

Fig. 2a: Why draw a line as guide to the eye and not the expected line for rotational diffusion?

Supporting information:

Some flaws:

main part of the manuscript → main part of the manuscript
are already is already?

Figure S3: I do not see reference to figure S3b in the text and no caption for the panel. The dotted lines at the high temperature data in Fig. S3a is confusing. They seem to be no fits but connections of points, while the low temperature data are fitted by equation (4).

Reviewer #2 (Remarks to the Author):

In this manuscript (COMMSCHEM-23-0552) the authors present experiments and calculation of the mobility of PPh₃ on a graphitic surface. Studying the atomic scale dynamics of molecules on surfaces serves as an important test for our understanding of molecule-surface interactions and the comparison to calculation methods paves the way of improving calculation accuracy to a degree it actually becomes useful for predicting a wide range of surface phenomena. This paper supplies both measurements and a comparison with theory and as such is definitely important and well worth publishing.

While I received both the manuscript and the supplementary information documents, I didn't seem to get a link to the video of the trajectories (or couldn't find it) so can't comment on that. Regarding the manuscript and the supplementary information section I have some issues I do not understand or have some concerns about, as listed below.

- Figure 2. The low temperature data is strange and its analysis is even stranger to me. From the caption it seems that the broadenings are larger at 200K than they are at 270K, is this just a mislabelling of the caption? If it isn't then what is the point of doing an Arrhenius analysis if the broadening decreases when the temperature is rising? I also don't see how the two panels are consistent looking at the upper panel all the green markers are under the blue ones, yet when you look at the bottom panel and focus on the highest temperature (presumably 200K and 270K?) for some Q values they go up for others down with temperature.....
- Figure 2. If there is Q dependent data for 50K and 100K I would suggest adding that to the upper panel as well.
- Figure 2. The error bars of the NSE data point seem strange, why are they so big? do they represent a standard deviation calculation from multiple measurements or an uncertainty from a fit procedure? Perhaps I missed it but I couldn't find an explanation what they mean and how they were estimated. If they represent a standard deviation then I would expect the measurements to be much more scattered, could these be double checked please?
- Figure 3a, the data set at 307K looks nothing like the 270K data shown in figure 2, is it because there is a huge change in the actual measurement for such a small temperature change or is it the way its analysed? Alternatively I might be missing the point here but in which case a more detailed explanation we be helpful for me and perhaps other readers.
- Figure 3c, there aren't enough marker labels on the vertical scale to readout the values, the reader needs to be able to extract numerical values from the graph
- Figure 3c. I am surprised by the relatively small error in D_0 , I would suggest rechecking this. Usually the uncertainty in the slope (activation error) which is actually quite significant results in a huge uncertainty in D_0 , I could be wrong but again urge this to be double checked.
- Figure 4 and text. The comparison of the broadenings / ISF extracted from the measurements and those extracted from the force field simulations seems to be only partially presented. From what I understand Figure 4 in the manuscript shows results from the force field simulations alongside two experimental data sets (TOF and NSE – blue markers). I presume the measurements were performed at about the same temperature the calculations were performed i.e. at about 300K, is this the case? I couldn't see that information, it doesn't look like the 307K data shown in figure 3a why is that? is it a different type of analysis?

What troubles me even more is the fact that the comparison is only made up to Q values of 1 inverse

angstrom, whereas broadenings were extract from measurements up to double that range. Is there an intrinsic reason why analysing trajectories from force field data can not give these values or is it that they simply deviate too much and mess up the comparison. If it's the later this makes it all the more important to show a full comparison for all the data you have. My recommendation regarding the force field is to present all the comparisons you have between the forcefield simulations and measurements (full range of Q values, full range of temperatures and all the different coverages you have, it is mentioned you measured at 0.2ML but no data is shown for this). I understand some of these extra graphs might need to be in the supplementary information which is fine as long as important issues (like misfits) are clearly discussed in the main text.

If there are aspects of the data the force field model does not reproduce they must be emphasized and discussed, otherwise the reader can be mislead into thinking this useful (but also extremely crude) theoretical tool (force field simulations) is more accurate than it actually is.

Other more minor points I noticed:

- Table 1. When discussing the DFT results some discussion of the expected accuracy (or functional dependency) would be good, it would be helpful if the reader could take this into consideration when comparing very small changes in the energies of different configurations.
- Page 2. When discussing the barrier to translation obtained from DFT, am I correct that it is assuming a straight motion between sites? If yes would it not decrease if the molecule is allowed to simultaneously rotate? I would think there might even be a particular rotation which could remove that barrier all together, am I wrong? It is kind of suggested in the last paragraph of the conclusions. If rotation reduces the barrier why would we expect the barrier for straight motion to fit the experimental value. I am confused, sorry....
- Page 3- first paragraph. Base temperature is not a well-defined temperature.
- Page 4 – the term a_{gr} is used, I imagine it is the lattice constant of graphite but this needs defining.
- Supplementary information – is there a reason / justification why the hydrogen is ignored in the ISF calculation? What happens if you include it? If it changes the fit then again this needs to be included in the paper and in which case this is not a minor point.
- Supplementary information. The graphs are titled "TPP", this is inconsistent with the rest of the paper.

Reviewer #3 (Remarks to the Author):

The paper provides a valuable examination of the dynamics of triphenylphosphine adsorbed on exfoliated graphite using neutron scattering techniques complemented by the DFT method and MD simulations. The studies prove to be interesting and significant, both in terms of fundamental understanding and potential applications. However, certain aspects require clarification, completion, and/or further development before the publication process.

1. One significant limitation in the work is the absence of a thorough comparison and discussion of

the results obtained for different samples (0.2, 0.5, and 0.9 ML). Although the authors examined three samples with varying layer coverage, there is a noticeable lack of analysis regarding the dynamics of these samples. Specifically, the following issues should be addressed:

- For low-temperature motion – how the activation energy (E_a) varies among the different samples.

- In the case of high-temperature motion – extend the discussion on how diffusion coefficient (D), relaxation time (τ), and length depend on the sample. Including a table with these parameters would offer valuable information to the reader.

Additionally, consider introducing further discussions about potential differences in the obtained parameters characterizing the motion of PPh₃. This would enhance the comprehensiveness of the study.

2. The low-temperature motion.

The authors discuss the Q -dependence of the width of the Lorentzian function, characterizing it as "rather flat." I can't fully agree with this statement, e.g. at 200 K, the width undergoes a change from 0.2 to approximately 0.42. Why the width dependence is not show up to $Q=2.0\text{\AA}^{-1}$ (as in fig. 3)?

Additionally, is it activation energy or diffusion barrier (as indicated in equation 3)?

Furthermore, the description of the model of "rotational motions" is not sufficient.

Is it the rotation of the molecule around an axis passing through the phosphorus atom and perpendicular to the surface of the layer? And how do we understand this motion in the frame of eq. 4, which is originally described as "one proton moving on the surface of a sphere" (ref. 26).

Why was the EISF not extracted from TOF data? I'm not convinced by the explanation that "While the uncertainty of the broadening in the temperature regime below about 300 K is relatively high, further evidence for the described type of motion comes from the elastic amplitude y_0 of the NSE data...". The uncertainty of the Lorentzian width (and broadening) is much bigger for NSE than for TOF data (fig. 2a). Even if with a quite large error, the EISF would be up to $Q = 2\text{\AA}^{-1}$, and not only up to 0.7\AA^{-1} , as for the elastic component data obtained from NSE (Fig. 3Sa).

Eq 4 and Fig 3S – If I understand correctly, the authors present elastic contribution of NSE signal using Equation 4 to describe rotational motion up to 350 K. The model for this motion remains unchanged with temperature, including higher temperatures (420 and 500 K) as shown in the figure

3S. However, we can read in the text: "rotation and motion of the phenyl groups are apparent at 50 – 300 K." This inconsistency needs clarification.

What does "A" represent in equation 4? The contribution of hydrogens(deuteriums) to the elastic line?

Furthermore, how should we interpret the statement "an increase of rotational dynamics"? Are the distinct fractions of H(D) observed in the TOF or NSE time window?

Are the dotted lines at 420 K and 500 K guided by eye?

As the width of Lorentzian function only slightly changes with temperature, it raises a question about the visibility of this motion at higher temperatures. Is it indeed a "constant offset" as suggested by the authors? Moreover, how does this contribution vary across different samples?

3. The nature of dynamics shifts from rotational motions to diffusive motions between 300-350 K.

What could be a possible reason for this change? Some discussion is needed.

4. Sample preparation. Has the procedure been previously validated? Were the samples preliminarily characterized, and how did the authors confirm the specified ML coverage?

5. Instrumental details. I would recommend adding a brief comment explaining why hydrogenated and deuterated samples were measured in TOF and NSE experiments, respectively. Furthermore, a short note that the motion of PPh₃ (not graphite) is observable in neutron experiments could be added.

6. Molecular dynamics simulations.

Any periodic conditions were applied? Or simulations were performed in a vacuum?

Any special reason to perform simulations only for 0.5 ML?

The snapshot of simulated system will be useful for the reader.

The simulations were conducted at four temperatures: 50 K, 150 K, 300 K, and 500 K. However, in the analysis, only data from 300 K were used, revealing a good agreement between simulated and experimental data. But, why is this comparison (as depicted in Fig. 4) presented only up to $Q=1.1 \text{ \AA}^{-1}$ and not extended to 2.0 \AA^{-1} ?

Furthermore, it would be interesting to explore a comparison between simulated data for low temperatures (50K or 150K, representing fully rotational dynamics) and high temperatures (500K, only translational motion) with experimental results. Additionally, extracting and comparing other

parameters such as activation energy (E_a), length, residence time and diffusion coefficient (D) from the simulations with experimental data could provide valuable insights.

6. I do not fully understand why the authors present animation only for one PPh₃ molecule on the graphite layer. The situation is significantly far from real structure. Perhaps a more insightful approach would be to present a video depicting the simulation of 16 PPh₃ molecules, which were previously compared with the experimental data?

The minor corrections:

The authors should revise the figure descriptions and captions to clearly indicate which sample and spectrometer the presented data corresponds to.

For instance:

- Fig. 2 – no information to which sample the data are referred. To 0.2 ML as in Fig. 2b?
- Fig. 3c – D for which sample?

Others:

- the “ τ ” symbol (below eq. 5) should be explained
- the “ α ” symbol should be also explained (page 4, left column)
- on page 4, left column, the value of diffusion coefficients should be corrected from 10^9 to 10^{-9} nm^2s^{-1}
- fig. S3, in the figure caption, “NSE” should be included. Additionally, there is no description of the Fig. S3b panel, and the authors do not make any reference to this Figure S3b.

Referee # 1

In their manuscript “Mobility of a Nanoscopic Moonlander: Translations and Rotations of Triphenylphosphine on Graphite” the authors present combined neutron spin echo, time of flight and DFT study of the dynamics of PPh₃ molecules on a graphite surface. The theoretical and experimental results agree well and the conclusions drawn are supported by the data presented. Overall, this presents a solid study of 2D diffusion on surfaces. This results is very interesting and shows how molecular rotations present at the lower temperatures investigated turn into lateral diffusion at higher temperatures. As stated by the authors the work extend existing literature on planar adsorbents to more complex systems. In the introduction the authors underline the importance of their findings with respect to the surface diffusion but they do not say how their results measuring the diffusion on graphite surfaces are related to those cases. Overall, I judge the manuscript very positive and support publication in Communications Chemistry.

We thank the reviewer for the assessment of our manuscript and the recognition of our results.

Below some minor points, which the authors should clarify prior to publication:

Low temperature dynamics: The authors discuss in the text different surface coverages but data is only shown for 0.5 ML in the manuscript. Some of the other data is shown in the supporting material but not all coverages mentioned. The authors should show the data to support their statements.

Author reply:

While we already had a few data points at different coverages, we have made this clearer in the revised version of the manuscript. The 0.5 ML data is the most reliable one as it includes both small-Q NSE data and TOF data at larger Q-values. Based on suggestions by the other referees, we have now also included an analysis of the 0.2 ML TOF data and we also discuss possible coverage-dependent influences in the manuscript.

Fig. 2a: Why draw a line as guide to the eye and not the expected line for rotational diffusion?

Author reply:

We agree and have now instead added a line based on uniaxial rotations of the molecule. Please note that Fig. 2a has been updated as there was a problem with the label in the original data (see comment to Ref #2).

Supporting information:

Some flaws:

main part of the manuscript —> main part of the manuscript
are already is already?

Corrected

Figure S3: I do not see reference to figure S3b in the text and no caption for the panel. The dotted lines at the high temperature data in Fig. S3a is confusing. They seem to be no fits but connections of points, while the low temperature data are fitted by equation (4).

Author reply:

We have adapted Fig. S3a, and due to a more extended discussion, we now include it as Fig 3a in the main manuscript. We have also included the 0.9 ML data as Fig. S3b. We now show the fit to all data, but note that at higher temperature, the translational motion starts to dominate the motion, and the fit is no longer such a good description of the data as also evident in the plots.

Referee # 2

Comments to the Author

In this manuscript (COMMSCHEM-23-0552) the authors present experiments and calculation of the mobility of PPh₃ on a graphitic surface. Studying the atomic scale dynamics of molecules on surfaces serves as an important test for our understanding of molecule-surface interactions and the comparison to calculation methods paves the way of improving calculation accuracy to a degree it actually becomes useful for predicting a wide range of surface phenomena. This paper supplies both measurements and a comparison with theory and as such is definitely important and well worth publishing.

Author reply:

We are pleased that the referee appreciates the impact of our results as well as the quality and originality of our work. We are grateful for the positive assessment and have made all the changes suggested by the referee, as detailed below.

While I received both the manuscript and the supplementary information documents, I didn't seem to get a link to the video of the trajectories (or couldn't find it) so can't comment on that.

Author reply:

The video was embedded in the Supplementary pdf but we realise that there seem to be issues depending on the pdf reader. We have now submitted the videos as separate files.

Regarding the manuscript and the supplementary information section I have some issues I do not understand or have some concerns about, as listed below.

- Figure 2. The low temperature data is strange and its analysis is even stranger to me. From the caption it seems that the broadenings are larger at 200 K than they are at 270 K, is this just a mislabelling of the caption? If it isn't then what is the point of doing an Arrhenius analysis if the broadening decreases when the temperature is rising? I also don't see how the two panels are consistent looking at the upper panel all the green markers are under the blue ones, yet when you look at the bottom panel and focus on the highest temperature (presumably 200 K and 270 K?) for some Q values they go up for others down with temperature

Author reply:

There was indeed a problem with the label and the associated plot of the data. This is due to the fact that the data was collected at different temperatures, both in the NSE and TOF experiments, due to limited time on the setup. We would like to apologise for the confusion this may have caused.

However, in the updated manuscript, we corrected the plot in Figure 2 and we can confirm that the broadening decreases - from 270 K to 300 K. We have therefore added an extended discussion to make our point clearer, including the intensity plots in the main text - Figure 3(a) which was previously in the supplement and Figure 3(b) which is completely new.

We believe that the origin of the unusual temperature dependence of the high-temperature broadening has to do with a combination of translations and rotation or perpendicular motion (possibly associated with hydrogens in the phenyl groups), both causing a broadening of similar strengths but with different levels of thermal activation - the onset of translational motion at higher temperatures means that both the rate and the broadening become dominated by the jump diffusion above 270 K - see added paragraphs in the "High temperature motion" section.

Finally, the main purpose of the Arrhenius plot is to show that (within the experimental uncertainties) there is hardly any activation barrier in the temperature range ≤ 300 K. In other words, it puts an upper limit to the activation energy for diffusion, as there is indeed variation in the data because of the uncertainties. Based on the other comments, we have now replaced the plot including data for different coverages - which also shows the variation of the individually determined activation energies E_a . We have also added some text - see also the next comment - and finally we confirm the activation energy also from the MD simulations.

- Figure 2. If there is Q dependent data for 50 K and 100 K, I would suggest adding that to the upper panel as well.

Author reply:

While we do have Q-dependent data for the other temperatures, the uncertainties of the fitted broadening are quite large meaning that they are pretty much scattered if plotted versus Q. Essentially they do not add

much meaningful information. This is mostly as the fitting of the broadening to the TOF data is pretty unreliable and gives rise to very large uncertainties. Nevertheless, for reference we have now added a plot in the supplementary information as Fig S3(a). Plus we have added a paragraph in the manuscript: “We note that the uncertainty of the broadening extracted from the neutron TOF data in the low-temperature regime below 300 K is relatively high, as shown in Figure S3(c) in the SI. Consequently, the data is quite scattered over the Q-range and there is also some variation in the E_a values calculated at different Q in Figure 2(b) meaning that $E_a \approx 5$ meV can be seen as an upper limit to the low-temperature motion.”

We have also added a new plot with the quasi-elastic intensities as Fig 3(b) in the revised manuscript which further adds to the discussion and understanding.

- Figure 2. The error bars of the NSE data point seem strange, why are they so big? do they represent a standard deviation calculation from multiple measurements or an uncertainty from a fit procedure? Perhaps I missed it but I couldn't find an explanation of what they mean and how they were estimated. If they represent a standard deviation then I would expect the measurements to be much more scattered, could these be double checked please?

Author reply:

As mentioned above there was a problem with the label and the plotted data - and we would like to apologise again for that. Nevertheless, the uncertainty of the low temperature NSE remains that large, at least in the low-temperature region. Exemplary fits to the NSE data are shown in Fig. S2 of the SI - which are weighted fits of the KWW function to the experimental data. The error bars/uncertainties correspond to 1σ confidence bounds. We have also added a line at the end of the Experimental measurements paragraph (page 3) regarding that.

- Figure 3a, the data set at 307K looks nothing like the 270K data shown in figure 2, is it because there is a huge change in the actual measurement for such a small temperature change or is it the way its analysed? Alternatively I might be missing the point here but in which case a more detailed explanation would be helpful for me and perhaps other readers.

Author reply:

Again, this is an unfortunate consequence of the initially wrong label and the problem should be resolved in the revised version of the manuscript.

- Figure 3c, there aren't enough marker labels on the vertical scale to readout the values, the reader needs to be able to extract numerical values from the graph

Author reply:

Has been corrected. We have now also included the 0.2 ML TOF data as requested by one of the other referees in the same plot.

- Figure 3c. I am surprised by the relatively small error in D_0 , I would suggest rechecking this. Usually the uncertainty in the slope (activation error) which is actually quite significant results in a huge uncertainty in D_0 , I could be wrong but again urge this to be double checked.

Author reply:

We are well aware that an extrapolation of the data to extract D_0 may be prone to large uncertainties, in particular if only a limited temperature window has been measured. D_0 is obtained from a weighted linear fit of the data and the uncertainty is almost 50% for the 0.5 ML data. For the new, added 0.2 ML data it is even larger (75%), which is due to the fact that we don't have reliable low-Q data (i.e. NSE data) at 0.2 ML.

- Figure 4 and text. The comparison of the broadenings / ISF extracted from the measurements and those extracted from the force field simulations seems to be only partially presented. From what I understand Figure 4 in the manuscript shows results from the force field simulations alongside two experimental data sets (TOF and NSE – blue markers). I presume the measurements were performed at about the same temperature the calculations were performed i.e. at about 300K, is this the case? I couldn't see that information, it doesn't look like the 307K data shown in figure 3a why is that? is it a different type of analysis?

What troubles me even more is the fact that the comparison is only made up to Q values of 1 inverse angstrom, whereas broadenings were extract from measurements up to double that range. Is there an

intrinsic reason why analysing trajectories from force field data can not give these values or is it that they simply deviate too much and mess up the comparison. If it's the later this makes it all the more important to show a full comparison for all the data you have. My recommendation regarding the force field is to present all the comparisons you have between the forcefield simulations and measurements (full range of Q values, full range of temperatures and all the different coverages you have, it is mentioned you measured at 0.2ML but no data is shown for this). I understand some of these extra graphs might need to be in the supplementary information which is fine as long as important issues (like misfits) are clearly discussed in the main text.

If there are aspects of the data the force field model does not reproduce they must be emphasized and discussed, otherwise the reader can be misled into thinking this useful (but also extremely crude) theoretical tool (force field simulations) is more accurate than it actually is.

Author reply:

We have replaced the figure of the MD simulations, showing the whole Q-range and a comparison with the 500 K measurements (n.b. as it is the only temperature where we have both NSE and TOF data as well as an MD simulation at the same - 0.5 ML - coverage. We discuss the discrepancy with respect to the experimental data and give possible reasons for that on page 5 as well as in the SI.

As suggested, we now show the results of the MD simulations up to the full Q range corresponding to the experimental measurements and discuss the discrepancies between the simulations and the measurements in the text. As previously described in the manuscript, the agreement between the simulations and the experiments is remarkable between $Q = 0$ and $Q = 1.3$. For $1.4 < Q < 2.0$, the simulated gamma continues to increase up to about 0.8 meV, while the measurements display a dip, with gamma slowly decreasing from 0.4 meV down to about 0.3 from $Q \geq 1.6$. Indeed, some divergences between the simulations and the experimental data were to be expected as we employed unconstrained force field MD without fitting any of the force field parameters. As the reviewer correctly points out, force field simulations provide a rather crude description of surface systems, and it is useful to highlight the weaknesses of MD in the paper. There are several possible reasons for the failure of MD simulations to describe gamma over the full range of Q values probed by the experiments, in particular for $Q > 1.3 \text{ \AA}^{-1}$. Furthermore, "universal" force fields such as COMPASS III are known to underestimate long-range dispersion forces. In this case, it appears that the force field does a relatively good job in capturing the dynamics assessed by the measurements at low to medium Q ($0 < Q < 1.3$) but fails to model the effect of the surface on short range motion in real space. We have now commented on the limits of MD simulations on page 5. The same force field was employed for simulating previous measurements on graphite for molecules with a different i.e. planar geometry showing good agreement and thus it is interesting and informative to comment on its limitations.

Other more minor points I noticed:

- Table 1. When discussing the DFT results some discussion of the expected accuracy (or functional dependency) would be good, it would be helpful if the reader could take this into consideration when comparing very small changes in the energies of different configurations.

Author reply:

We thank the reviewer for the insightful comment. Indeed, the accuracy of density functional theory calculations has been a topic of importance and debate in the last three decades. Hundreds of benchmark studies exist where the performance of exchange-correlation functionals, for thousands of systems, are compared. In general, it is accepted that a functional performance within a particular class of molecules is not a guarantee that the same functional will perform equally well with similar systems. In the past twelve years, this and other groups assessed and compared the performance of several XC functionals and dispersion corrections in the context of surface diffusion and dynamics[1-6]. Comparison with QENS and HeSE measurements consistently showed that PBE with TS long-range dispersion corrections performs well and with comparable accuracy for all the systems investigated, with typical errors in the meV range.

1. Lechner, B. A. J.; Hedgeland, H.; Ellis, J.; Allison, W.; Sacchi, M.; Jenkins, S. J.; Hinch, B. J. Quantum Influences in the Diffusive Motion of Pyrrole on Cu(111). *Angew. Chem. Int. Ed.* 2013, 52, 5085–5088.
2. Lechner, B. A. J.; Sacchi, M.; Jardine, A. P.; Hedgeland, H.; Allison, W.; Ellis, J.; Jenkins, S. J.; Dastoor, P. C.; Hinch, B. J. Jumping, Rotating, and Flapping: The Atomic-Scale Motion of Thiophene on Cu(111). *J. Phys. Chem. Lett.* 2013, 4, 1953–1958,
3. Hedgeland, H.; Sacchi, M.; Singh, P.; McIntosh, A. J.; Jardine, A. P.; Alexandrowicz, G.; Ward, D. J.; Jenkins, S. J.; Allison, W.; Ellis, J. Mass Transport in Surface Diffusion of van der Waals Bonded Systems: Boosted by Rotations? *J. Phys. Chem. Lett.* 2016, 7, 4819–4824.

4. Calvo-Almazán, I.; Sacchi, M.; Tamtögl, A.; Bahn, E.; Koza, M. M.; Miret-Artés, S.; Fouquet, P. Ballistic diffusion in poly-aromatic hydrocarbons on graphite. *J. Phys. Chem. Lett.* 2016, 7, 5285–5290.
5. Sacchi, M.; Singh, P.; Chisnall, D. M.; Ward, D. J.; Jardine, A. P.; Allison, W.; Ellis, J.; Hedgeland, H. The dynamics of benzene on Cu(111): a combined helium spin echo and dispersion-corrected DFT study into the diffusion of physisorbed aromatics on metal surfaces. *Faraday Discussions* 2017, 204, 471–485.
6. Maier, P.; Xavier, N. F.; Truscott, C. L.; Hansen, T.; Fouquet, P.; Sacchi, M.; Tamtögl, A. How does tuning the van der Waals bonding strength affect adsorbate structure? *Phys. Chem. Chem. Phys.* 2022, 24, 29371–29380.

In this context, we added the following paragraph on page 2 of the revised manuscript:

”Given the chemical similarity between PPh₃ and previously reported organic precursors, we expect that the accuracy of our calculations with PBE and TS corrections will be comparable with our previous studies on graphitic and similar surfaces [4,5,20,21] on the order of 10 meV.”

• Page 2. When discussing the barrier to translation obtained from DFT, am I correct that it is assuming a straight motion between sites? If yes would it not decrease if the molecule is allowed to simultaneously rotate? I would think there might even be a particular rotation which could remove that barrier all together, am I wrong? It is kind of suggested in the last paragraph of the conclusions. If rotation reduces the barrier why would we expect the barrier for straight motion to fit the experimental value. I am confused, sorry....

Author reply:

The reviewer is correct in assuming that we approximate the jump-diffusion events between neighbouring sites as a straight motion and we have therefore added that in the text on page 5.

This assumption is justified by our experience in simulating surface dynamics events on several other surface systems, including graphitic surface, where we observed that, in very good approximation, the internal degrees of freedom of the molecular adsorbate are essentially “frozen” or uncoupled to the generalised reaction coordinate describing the diffusion event.

Even if we allowed the molecule to rotate around the C3 axis in the "5" position, the barrier would not have decreased.

• Page 3- first paragraph. Base temperature is not a well-defined temperature.

Author reply:

We agree and have corrected it to ≈ 2 K

• Page 4 – the term a_{gr} is used, I imagine it is the lattice constant of graphite but this needs defining.

Author reply:

Correct and has been added.

• Supplementary information – is there a reason / justification why the hydrogen is ignored in the ISF calculation? What happens if you include it? If it changes the fit then again this needs to be included in the paper and in which case this is not a minor point.

Author reply:

There seems to be a misunderstanding: The hydrogen atoms are not ignored in the calculation of the ISFs. The so-called “total” ISF is calculated from the trajectories of all atoms, weighted with the corresponding neutron scattering cross sections. The latter means that the H-atoms will contribute the most to the signal and in fact the ISFs are almost identical if one calculates the ISF from the H-atoms alone compared to the total ISF. The corresponding information can be found in the SI and we refer to that in the main text.

• Supplementary information. The graphs are titled “TPP”, this is inconsistent with the rest of the paper.

Author reply:

We agree and have corrected the figure accordingly.

Referee # 3

The paper provides a valuable examination of the dynamics of triphenylphosphine adsorbed on exfoliated graphite using neutron scattering techniques complemented by the DFT method and MD simulations. The studies prove to be interesting and significant, both in terms of fundamental understanding and potential applications.

We thank the referee for the positive assessment and have made all the changes suggested by the referee, as detailed below.

However, certain aspects require clarification, completion, and/or further development before the publication process.

- 1) One significant limitation in the work is the absence of a thorough comparison and discussion of the results obtained for different samples (0.2, 0.5, and 0.9 ML). Although the authors examined three samples with varying layer coverage, there is a noticeable lack of analysis regarding the dynamics of these samples. Specifically, the following issues should be addressed:

For low-temperature motion – how the activation energy (E_a) varies among the different samples.

In the case of high-temperature motion – extend the discussion on how diffusion coefficient (D), relaxation time (τ), and length depend on the sample. Including a table with these parameters would offer valuable information to the reader.

Additionally, consider introducing further discussions about potential differences in the obtained parameters characterizing the motion of PPh₃. This would enhance the comprehensiveness of the study.

Author reply:

Based on the reviewer's suggestions, we have adapted the manuscript and included more data for the three different coverages. The 0.5 ML coverage allows us still to draw the most reliable conclusions as we have a combination of low Q NSE data and TOF data at larger Q -values. Nevertheless, we have now also added an analysis of the 0.2 ML TOF data (in a revised version of Fig 3(c)) while the table itself is in the SI to not further extend the length of the manuscript). Similarly, for the 0.9 ML NSE data, we added a Figure in the SI. We further discuss possible coverage effects in a separate paragraph before the conclusions as well as in the SI.

- 2) The low-temperature motion.

The authors discuss the Q -dependence of the width of the Lorentzian function, characterizing it as "rather flat." I can't fully agree with this statement, e.g. at 200 K, the width undergoes a change from 0.2 to approximately 0.42. Why the width dependence is not shown up to $Q=2.0\text{\AA}^{-1}$ (as in fig. 3)? Additionally, is it activation energy or diffusion barrier (as indicated in equation 3)?

Author reply:

As mentioned in our reply to Ref #2, there was a problem with the label and we have corrected the plot in Figure 2, showing now also the dependence up to 2 Angstrom⁻¹. We have removed the statement and included a more detailed discussion of the Q -dependence in the low-temperature regime on page 3.

We agree that the term diffusion barrier is unfortunate - in particular in the low-temperature regime and have replaced it with "activation energy"

Furthermore, the description of the model of "rotational motions" is not sufficient. Is it the rotation of the molecule around an axis passing through the phosphorus atom and perpendicular to the surface of the layer? And how do we understand this motion in the frame of eq. 4, which is originally described as "one proton moving on the surface of a sphere" (ref. 26).

Author reply:

As mentioned above, we have now added a more detailed discussion of the Q -dependence in the low-temperature regime on page 3 - that includes references to previous measurements of small molecules on graphite. We have also added a comment to the manuscript that it is of course a very simplified model, but without better experimental data and other analytic solutions for a complex molecule such as PPh₃ we believe that it is adequate in the present case.

Why was the EISF not extracted from TOF data? I'm not convinced by the explanation that "While the uncertainty of the broadening in the temperature regime below about 300 K is relatively high, further evidence for the described type of motion comes from the elastic amplitude y_0 of the NSE data...". The uncertainty of the Lorentzian width (and broadening) is much bigger for NSE than for TOF data (fig. 2a). Even if with a quite large error, the EISF would be up to $Q = 2A^{-1}$, and not only up to $0.7 A^{-1}$, as for the elastic component data obtained from NSE (Fig. 3Sa).

Author reply:

We have added the intensity plots in the main text as Figure 3(a) and a plot of the quasi-elastic intensities from the TOF measurements in Fig. 3(b). Nevertheless, the y_0 plots from the NSE exhibit much smaller uncertainties, as for the EISF in the low temperature TOF measurements we rely on 3 fitting parameters (width and height of the Lorentzian plus height of the elastic peak). Therefore, we have added the following line in the manuscript:

"Moreover, further evidence for the described type of motion comes from the NSE data. As it is difficult to extract the actual elastic incoherent structure factor (EISF) due to the uncertainties of the fits in the low temperature regime, we consider instead the elastic amplitude y_0 of the NSE data according to Equation 2"

Eq 4 and Fig 3S – If I understand correctly, the authors present elastic contribution of NSE signal using Equation 4 to describe rotational motion up to 350 K. The model for this motion remains unchanged with temperature, including higher temperatures (420 and 500 K), as shown in the Figure S3. However, we can read in the text: "rotation and motion of the phenyl groups are apparent at 50 –300 K." This inconsistency needs clarification.

Author reply:

We changed the text to make clear that up to about 350 K the rotational / "flapping" motion dominates while above it is translational motion.

That goes in hand with a change of the NSE signal, which becomes difficult to fit to Equation 4 above 350 K, while the overall change to the motion is then better seen in plots of the broadening versus the entire Q-range, i.e. including the TOF data.

What does "A" represent in equation 4? The contribution of hydrogens(deuteriums) to the elastic line?

Author reply:

A is the purely elastic signal in the NSE measurements, e.g., the contribution of the carbon substrate. We have also added this in the manuscript.

Furthermore, how should we interpret the statement "an increase of rotational dynamics"? Are the distinct fractions of H(D) observed in the TOF or NSE time window? Are the dotted lines at 420 K and 500 K guided by eye?

Author reply:

We have changed the text to "increasing contribution" and have replaced all lines with fits, but note that at higher temperatures, the translational motion starts to dominate over the rotational and flapping motion and consequently the fit is no longer a good description in that temperature regime.

As the width of Lorentzian function only slightly changes with temperature, it raises a question about the visibility of this motion at higher temperatures. Is it indeed a "constant offset" as suggested by the authors? Moreover, how does this contribution vary across different samples?

Author reply:

As described in the discussion - see the answer to point 3 - both processes occur concurrently but with different levels of thermal activation, meaning that in the high-temperature regime a constant offset is maintained while the signature from the jump-diffusion dominates the Gamma-Q dependence. We have now added a discussion about variation over the different coverages and refer here also to possible effects on this type of motion.

- 3) The nature of dynamics shifts from rotational motions to diffusive motions between 300-350 K. What could be a possible reason for this change? Some discussion is needed.

Author reply:

We have added a discussion about the changes on page 4 in the “High-temperature motion” section and as mentioned above we have also added a plot with the quasi-elastic intensities over the entire probed temperature region to make that clearer.

- 4) Sample preparation. Has the procedure been previously validated? Were the samples preliminarily characterized, and how did the authors confirm the specified ML coverage?

Author reply:

The procedure has been previously validated and we have added two references in the experimental details section. Calibration for ML coverage comes from published molecularly resolved scanning tunnelling microscopy data as also cited in the Sample preparation section.

- 5) Instrumental details. I would recommend adding a brief comment explaining why hydrogenated and deuterated samples were measured in TOF and NSE experiments, respectively. Furthermore, a short note that the motion of PPh3 (not graphite) is observable in neutron experiments could be added.

Author reply:

We have added a paragraph at the end of the Experimental section (page 3) as well as two sentences in the instrumental details section (page 7)

- 6) Molecular dynamics simulations.

Any periodic conditions were applied? Or simulations were performed in a vacuum? Any special reason to perform simulations only for 0.5 ML?

The snapshot of the simulated system will be useful for the reader.

Author reply:

The simulations were performed with periodic boundary conditions. We have added some details, including also a snapshot of the simulated system in the supplementary information. We chose to simulate the 0.5 ML setup, as we do have most experimental data (i.e. both NSE and TOF data) for that coverage.

The simulations were conducted at four temperatures: 50 K, 150 K, 300 K, and 500 K. However, in the analysis, only data from 300 K were used, revealing a good agreement between simulated and experimental data. But, why is this comparison (as depicted in Fig. 4) presented only up to $Q=1.1 \text{ \AA}^{-1}$ and not extended to 2.0 \AA^{-1} ?

Author reply:

We have replaced the figure of the MD simulations, showing the whole Q-range and a comparison with the 500 K measurements - the latter is (unfortunately) the only temperature where we have both NSE and TOF data as well as an MD simulation at the same (0.5 ML) coverage.

In that context we have also added an extended discussion on page 5.

Furthermore, it would be interesting to explore a comparison between simulated data for low temperatures (50K or 150K, representing fully rotational dynamics) and high temperatures (500K, only translational motion) with experimental results. Additionally, extracting and comparing other parameters such as activation energy (E_a), length, residence time and diffusion coefficient (D) from the simulations with experimental data could provide valuable insights.

Author reply:

We have extended the entire MD discussion and we now also extract the activation energy for the low-temperature regime from the MD simulations. While we did analyse the trajectories of all 4 temperatures for which MD simulations were performed, and we agree that an even more detailed analysis of the MD simulation, including residence times, would be interesting, but we would like to note that it goes beyond the current study.

Such a detailed analysis of MD simulations could be a separate manuscript on itself and would also exceed the expected length of articles in the journal - yet we are no experts on MD simulations and it would possibly require a different approach to the here used forcefield MD simulations with a commercial software (Material Studio). The main purpose of the MD simulations for the manuscript is to get a “real-space illustration” of the motion and to aid the interpretation of the experimental data. We

do of course appreciate that more valuable insights could be obtained, and we hope that our manuscript will encourage possible further research down that line.

- 7) I do not fully understand why the authors present animation only for one PPh3 molecule on the graphite layer. The situation is significantly far from the real structure. Perhaps a more insightful approach would be to present a video depicting the simulation of 13 PPh3 molecules, which were previously compared with the experimental data?

Author reply:

We agree and have now included two videos of the MD simulation from the 0.5 ML setup. On the other hand, the simulation for a single molecule illustrates the individual motion in a better way, which is why we have also kept that video.

The minor corrections:

The authors should revise the figure descriptions and captions to clearly indicate which sample and spectrometer the presented data corresponds to.

For instance:

- Fig. 2 – no information to which sample the data are referred. To 0.2 ML as in Fig. 2b?
- Fig. 3c – D for which sample?

Author reply:

Corrected.

Others:

- the “tau” symbol (below eq. 5) should be explained
- We have added below eq. 5 that tau is the residence time, we have also changed the tau obtained from the ISF to τ_{ww} to avoid any possible confusion.
- the “agr” symbol should be also explained (page 4, left column)
- We have added a comment, that a_{gr} is the graphite in-plane lattice constant.
- on page 4, left column, the value of diffusion coefficients should be corrected from 10^9 to 10^{-9} nm^2s^{-1}
- Corrected.
- fig. S3, in the figure caption, "NSE" should be included. Additionally, there is no description of the Fig. S3b panel, and the authors do not make any reference to this Figure S3b.
- Corrected. Fig S3(a) has now been moved to the main text, and instead as Fig S3(a) we shows two data sets taken at low temperature which is now also referred to in the main text and in Fig S3(b) the high coverage (0.9 ML) NSE data has been added.

REVIEWERS' COMMENTS:

Reviewer #2 (Remarks to the Author):

[Editorial Note: No further comments for the authors.]

Reviewer #3 (Remarks to the Author):

The authors have diligently addressed queries and refined their work in accordance with suggestions provided. From my perspective, the work is primed for publication, with only minor editorial observations:

1. I would suggest to add in the abstract information about the temperature range - what type of motions are visible at low and at high temperature
2. On page 5, in the left column, the first row displays "1.1 – 1.3 agr." Is this correct, or should it be "(1.1 – 1.3) agr"?
3. In certain instances (e.g., Figure 4 captions), the authors reference "(5)" when it should likely be referred to as "eq. 5".
4. Were MD simulations conducted for 3 graphite layers and 13 PPh3 molecules in vacuum? Were periodic boundary conditions applied for the system?
5. Additionally, the description of the video showing a single PPh3 molecule is not clear for me (in SI is written "PPh3atom"). I mean - was the simulation performed for a single molecule? Or for 13 molecules, and one was extracted and presented in the video? Please, clarify in the text.

Reviewer #3 (Remarks to the Author):

The authors have diligently addressed queries and refined their work in accordance with suggestions provided. From my perspective, the work is primed for publication, with only minor editorial observations:

1. I would suggest to add in the abstract information about the temperature range - what type of motions

are visible at low and at high temperature:

We have added the following sentence:

While molecular rotations dominate up to about 300 K, the molecules follow an additional translational jump-motion across the surface from 350-500 K

2. On page 5, in the left column, the first row displays "1.1 – 1.3 agr." Is this correct, or should it be "(1.1 –

1.3) agr"?

The referee is correct and we have added the brackets accordingly.

3. In certain instances (e.g., Figure 4 captions), the authors reference "(5)" when it should likely be referred to as "eq. 5".

Has been corrected so that at all instances Equation is preceding the number

4. Were MD simulations conducted for 3 graphite layers and 13 PPh3 molecules in vacuum? Were periodic boundary conditions applied for the system?

Yes that's correct, and the details and the setup are given in the Supplementary Information. We have made a note to the SI in the Methods section and added an additional line at the instance in the SI to make that clearer.

5. Additionally, the description of the video showing a single PPh3 molecule is not clear for me (in SI is written "PPh3atom'). I mean - was the simulation performed for a single molecule? Or for 13 molecules, and one was extracted and presented in the video? Please, clarify in the text.

The latter was a setup with a single adsorbed PPh3-atom. We have added a short paragraph in the SI to explain that / make that clear to the reader.